# IL-33 enhances Jagged1 mediated NOTCH1 intracellular domain (NICD) deubiquitination and pathological angiogenesis in proliferative retinopathy

Deepti Sharma[1,2], Shivantika Bisen[1,2], Geetika Kaur[1,2], Eric C. Van Buren[3], Gadiparthi N. Rao[4] & Nikhlesh K. Singh [1,2✉]

Pathological retinal neovascularization (NV) is a clinical manifestation of various proliferative retinopathies, and treatment of NV using anti-VEGF therapies is not selective, as it also impairs normal retinal vascular growth and function. Here, we show that genetic deletion or siRNA-mediated downregulation of IL-33 reduces pathological NV in a murine model of oxygen-induced retinopathy (OIR) with no effect on the normal retinal repair. Furthermore, our fluorescent activated cell sorting (FACS) data reveals that the increase in IL-33 expression is in endothelial cells (ECs) of the hypoxic retina and conditional genetic deletion of IL-33 in retinal ECs reduces pathological NV. In vitro studies using human retinal micro-vascular endothelial cells (HRMVECs) show that IL-33 induces sprouting angiogenesis and requires NFkappaB-mediated Jagged1 expression and Notch1 activation. Our data also suggest that IL-33 enhances de-ubiquitination and stabilization of Notch1 intracellular domain via its interaction with BRCA1-associated protein 1 (BAP1) and Numb in HRMVECs and a murine model of OIR.

[1] Integrative Biosciences Center, Wayne State University, Detroit, MI 48202, USA. [2] Department of Ophthalmology, Visual and Anatomical Sciences, Wayne State University School of Medicine, Detroit, MI 48202, USA. [3] Department of Oncology, Wayne State University School of Medicine, Detroit, MI 48201, USA. [4] Department of Physiology, University of Tennessee Health Science Center, Memphis, TN 38163, USA. ✉email: nsingh2@wayne.edu

The retina consumes a high proportion of oxygen, polyunsaturated fatty acids and is susceptible to oxidative stress[1]. The retinal and choroidal vascular plexi regulate the elevated metabolic supply to the retina, and any abnormality or breakdown in these vascular plexi leads to tissue ischemia and pathological retinal neovascularization. Current treatment regimens for these neovascular diseases, such as laser photocoagulation and anti-VEGF therapies are effective but carry risk[2–4]. Laser photocoagulation therapies lead to the development of macular edema, visual field loss, and reduced night vision[5]. Anti-VEGF therapies require regular intravitreal injections, leading to infections and endophthalmitis[6]. In addition, anti-VEGF therapies lead to tractional retinal detachment and retinal photoreceptors atrophy[7]. Recent studies have shown that NV is governed by a complex interplay between immune cells and inflammatory cytokines[8]. Furthermore, it has also been reported that immune cells are recruited to the hypoxic tissue to induce the expression of cytokines to make a pro-angiogenic microenvironment[9]. The present understanding on the role of cytokines and inflammatory processes in the pathogenesis of retinal NV is at an early stage, and the mechanism(s) by which inflammatory processes regulate visual dysfunction needs to be addressed.

Notch signaling is dysregulated in numerous diseases[10,11]. The Notch signaling pathway consists of 5 ligands (Delta-like 1, 3, and 4, Jagged1 and 2) and 4 transmembrane receptors (Notch1- 4). The Notch signaling pathway is activated by the binding of a Notch ligand to a receptor, which leads to sequential cleavages of the receptors, leading to Notch intracellular domain (NICD) release. NICD translocates to the nucleus and induces the transcription of Notch target genes[12]. The Notch ligands, DLL4 and Jagged1 are shown to have a role in developmental angiogenesis. DLL4 is the most characterized ligand for Notch1 activation and is required for normal arterial development in an embryo[13]. DLL4 has been shown to inhibit wound healing and neoangiogenesis in adults[14]. Jagged1 is a less studied Notch ligand, required for the development of heart and head vasculature in an embryo[15]. Jagged1 promotes tip cell formation and sprouting angiogenesis in endothelial cells[16,17]. In addition, inhibition of Jagged1-Notch1 signaling has been shown to reduce angiogenesis and pericyte-endothelial cell interactions[18]. The role of Jagged1 in ischemic retinopathies has not yet been thoroughly studied, and further investigations are required to understand the involvement of Jagged1-regulated Notch pathway in ischemic retinopathies.

The Interleukin-1 (IL-1) family of cytokines are subdivided into three major groups: the IL-1, IL-18, and IL-36 subfamilies. The IL-1 subfamily of cytokines includes IL-1alpha, IL-1beta, and IL-33[19]. IL-1alpha and IL-1beta are widely studied IL-1 subfamily members and are associated with retinal degenerative diseases and angiogenesis with little information on IL-33[19]. IL-33 is expressed by epithelial cells[20], fibroblasts[20], and endothelial cells[21]. The role of IL-33 in angiogenesis is still not delineated, with the majority of reports showing a proangiogenic role for IL-33[22], while a few demonstrating its anti-angiogenic role[23]. IL-33 induces von Willebrand factor expression and angiogenesis in a concentration-dependent manner in human endothelial cells[24]. IL-33 also improves wound healing in diabetic mice by increasing extracellular matrix deposition and neovascularization[25]. In addition, increased expression of IL-33 enhances angiogenesis in HIF-1alpha dependent manner in hypoxic human pulmonary artery endothelial cells[26]. IL-33 is a signaling molecule that participates in processes as varied as inflammation, autoimmunity, organ fibrosis, and cardiac injury[20,27]. Therefore, genetic deletion studies are required to explore the effect of IL-33 on post-ischemic neo-angiogenesis.

Here, we looked for the role of IL-33 and the pathways it regulates to selectively target pathological neo-angiogenesis in hypoxic/ischemic retinopathies. We provide evidence that hypoxia/ischemia-induced IL-33 expression in endothelial cells regulates angiogenic sprouting and tufts formation in the retina and depletion of IL-33 levels in a murine model of oxygen-induced retinopathy (OIR) ameliorates retinal neovascularization. Here we identified the functional role of IL-33 induced Jagged1/Notch1 signaling in post-ischemic neo-angiogenesis. Our results demonstrate that Jagged1 not only induces Notch1 activation in retinal endothelial cells but also regulates its deubiquitylation and stabilization via its interaction with Numb and BAP1.

## Results

**IL-33 induces angiogenic effects in human retinal microvascular endothelial cells.** The current therapies to block angiogenesis are mostly focused on VEGF-A neutralization or blockade of VEGFR2 signaling. Although, several studies have shown that patients suffering from cancer or age-related macular degeneration (AMD) do not recuperate to anti-VEGF-A therapies[28,29]. Anti-VEGF-A therapies have been reported to cause tractional retinal detachments[7]. Recently, many studies have reported that neovascularization is governed by a complex interplay between immune cells and inflammatory cytokines[19]. IL-1 is a master regulator of inflammation, and the IL-1 subfamily of cytokines includes IL-1alpha, IL-1beta, and IL-33. IL-1alpha and IL-1beta are associated with retinal degenerative diseases and angiogenesis with little information on IL-33[20]. Therefore, we looked for the expression of IL-33 in a murine model of OIR (Fig. 1a). In OIR, mouse pups are first exposed to 75% oxygen (hyperoxia) for 5 days (P7–P12), which leads to regression of vessels and cessation of normal radial vessels, mimicking the first phase of retinopathy of prematurity (ROP). When the pups were returned to ambient air (normoxia) at P12, the avascular areas of the retina become hypoxic, leading to increased expression of various angiogenic factors, which results in excessive retinal neovascularization. The neovascular phase of this model depicts the second phase of ROP in humans and mimic certain symptoms of proliferative diabetic retinopathy. The retinas from various periods of relative hypoxia were analyzed for the expression of cytokines and interleukins.

We observed an increased expression of IL-33 both at mRNA and protein levels in hypoxic retinas compared to normoxia (Fig. 1b, c), which was maximum at 72 h, and then it came back to normal at 120 h. In the murine OIR model, maximum retinal neovascularization is seen at P17 (120 h post hypoxia), after which regression of neovascularization takes place. Therefore, our data suggest that IL-33 via Jagged1-Notch1 signaling regulate neovascularization at P17 with no effect on retinal repair or vessel regression. To understand the functional significance of IL-33 in the hypoxic retina, we next studied the effect of IL-33 on proliferation, migration, sprouting, and tube formation of HRMVECs. A dose-response study of IL-33 on angiogenic effects of HRMVECs showed that IL-33 has a maximal effect at 20 ng/mL (Supplementary Fig. 1). The thymidine incorporation assay and FITC BrdU flow cytometry assay were used to evaluate the effect of IL-33 on proliferation. We failed to observe any significant effect of IL-33 on HRMVECs proliferation (Fig. 1d, e). The effect of IL-33 on the migration of HRMVECs was observed using a modified Boyden Chamber Assay and wound healing assay. IL-33 increased the migration of HRMVECs at 20 ng/mL (Fig. 1f, g). The effect of IL-33 on tip cell formation/sprouting and tube formation was assessed using a 3-D angiogenesis assay and a 2-dimensional Matrigel assay, respectively. The IL-33 treatment at 20 ng/mL led to an increase in the number of sprouts/bead and extensive networks, compared with control-treated cells (Fig. 1h, i). IL-33 is a functional ligand to its receptor ST2. Therefore, we studied the role of ST2 in IL-33-induced angiogenic effects in

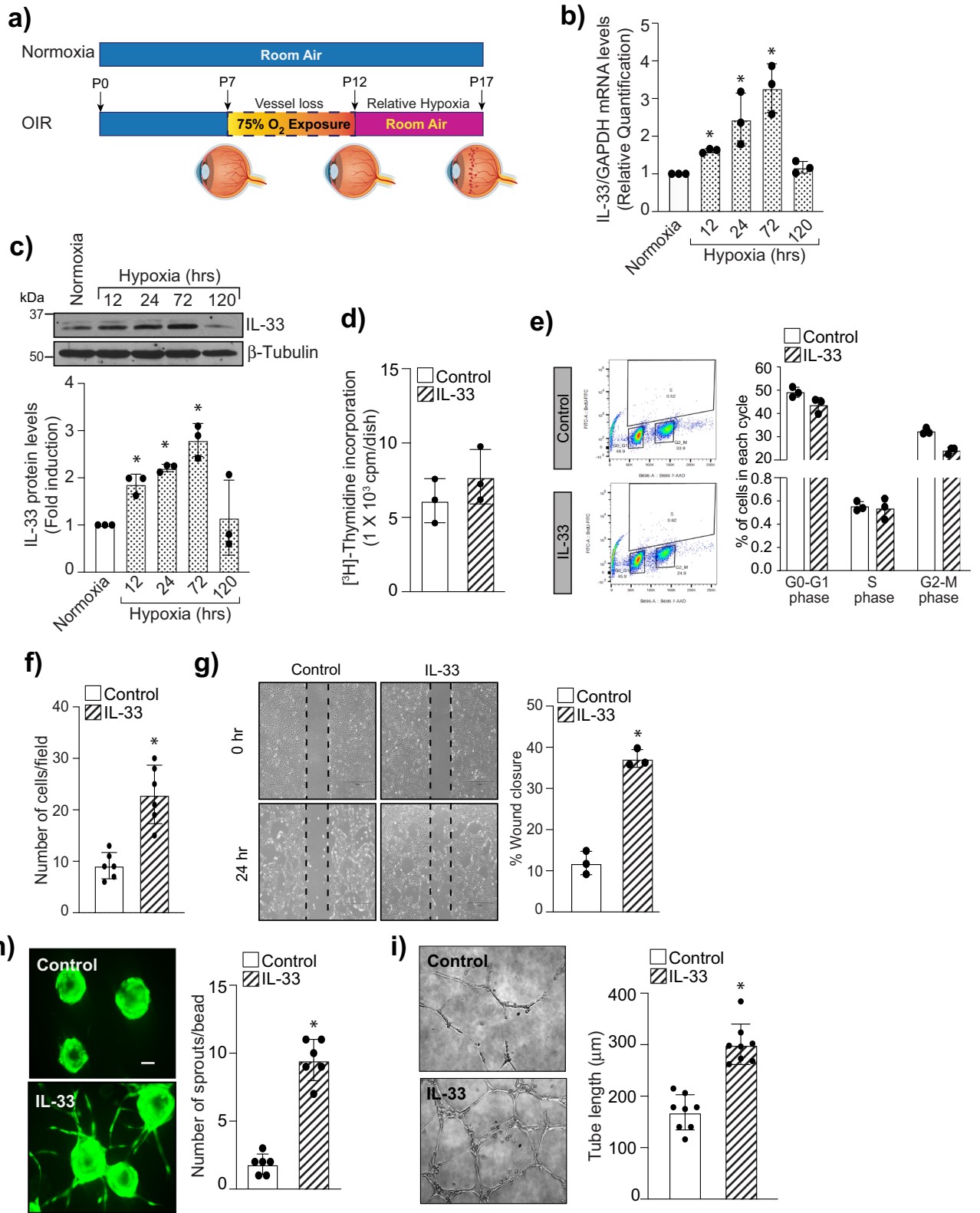

HRMVECs. Depletion of ST2 levels using its siRNA, significantly reduced IL-33-induced migration, tube formation, and sprouting of HRMVECs (Supplementary Fig. 2).

**IL-33 regulates pathological retinal neovascularization in a murine model of retinopathy.** Since we observed a role for IL-33

in HRMVECs migration, sprouting, and tube formation, we next looked for the effect of IL-33 on OIR-induced retinal neovascularization. We first assessed the effectiveness of intravitreal delivery of siRNA into vascular cells through intravitreal injections. We injected 0.5 µl of invivofectamine 2-siRNA duplex solution containing 0.6 µg of rhodamine-labeled non-targeted siRNA at P12 and P14. At P17 the retinal cross-sections were

**Fig. 1 IL-33 induce sprouting angiogenesis in retinal endothelial cells. a** Schematic diagram representing the murine OIR model. C57BL/6 mice pups with dams were exposed to 75% oxygen from P7 to P12 and returned to room at P12. At P17, eyes were enucleated, retinas isolated. **b** Total cellular RNA was isolated from retina and quantified for IL-33 and GAPDH mRNA levels by QRT-PCR. **c** Retinal tissue extracts were prepared and analyzed for IL-33 by Western blotting and normalized to β-tubulin. $n = 6$ mice per group. Quiesced HRMVECs were treated with and without IL-33 (20 ng/mL) and cell proliferation was measured by thymidine incorporation (**d**), and BrdU cell proliferation assay (**e**). **f, g** Everything is same as in **d**, except that cell migration was measured using Boyden chamber method (**f**) and wound healing assay (**g**). **h** HRMVECs cells were labeled, coated onto cytodex beads, embedded in a 3D-fibrin gel and sprouting was observed after 3 days under Zeiss LSM800 microscope. **i** Everything is same as in **d**, except that tube formation was assessed using growth factor reduced Matrigel. The bar graphs show the quantitative analysis of three independent experiments, expressed as Mean ± SD. *$P < 0.05$ vs control or normoxia. Scale bar represents 50 μm in **h**.

made and stained for CD31. We observed colocalization of CD31 with rhodamine, indicating the presence of siRNA molecules in vascular cells (Fig. 2a). We observed the presence of intravitreally injected siRNA molecules in other retinal cells too. The intravitreal injection of IL-33 siRNA reduced IL-33 levels in the whole retinal lysate (Fig. 2b). The downregulation of IL-33 had a protective effect against pathological neoangiogenesis. The retinas of mice subjected to the OIR model were stained with isolectin B4 (a marker of endothelial cells), flat-mounted, and quantified as detailed in "Methods". Downregulation of IL-33 levels attenuated hypoxia-induced retinal endothelial cell (EC) filopodia formation suggesting the role of IL-33 in endothelial tip cell formation (Fig. 2c). The pathological neovascularization was significantly reduced in mice pups treated with IL-33 siRNA compared to control siRNA group (Fig. 2d). No significant changes were observed in the vasoobliterated/total retina area in IL-33 siRNA group compared to control siRNA group (Fig. 2e). These findings suggest that downregulation of IL-33 levels in the hypoxic retina resulted in reduced neovascularization with no significant change in avascular area.

OIR is a well-characterized model in which increased VEGFA expression has been observed in the retina[30,31], and anti-VEGF-A therapies have been shown to inhibit neovascularization in several diseases[32]. Therefore, to assess whether IL-33 controls VEGFA expression or its' signaling in retinal endothelial cells, we tested the time course effect of IL-33 on VEGFA expression and VEGFR2 activation in HRMVECs. No significant changes in VEGFA expression or VEGFR2 activation were observed in IL-33 treated HRMVECs (Fig. 2f). We also failed to observe any significant changes in IL-33 levels in VEGF-A treated HRMVECs (Fig. 2g).

To further confirm our observations that depletion in IL-33 levels reduces neovascularization in the OIR model, we genetically deleted IL-33 in mouse pups and looked for its role in OIR-induced pathological retinal neovascularization (Fig. 3a). IL-33 deficient mice showed a complete reduction of IL-33 in the whole retinal lysate (Fig. 3b) and a reduced hypoxia-induced tip cell formation or sprouting angiogenesis (Fig. 3c) compared to IL-$33^{+/+}$ mice. A significant reduction in retinal neovascularization was observed in P17 flat-mounted retinas of IL-33 deficient mice (IL-$33^{-/-}$), showing a reduction of neovascular tufts formation, despite comparable avascular zones (Fig. 3d, e). The results shown above suggest that IL-33 can be used for treating pathological retinal neovascularization, as it reduces vascular overgrowth without having any effect on VEGFA expression. These findings can be very important in proliferative retinopathies treatment as VEGF-A is a survival factor for neurons[33].

**Endothelial specific expression of IL-33 regulates retinal neovascularization in the hypoxic retina.** IL-33 is considered an alarmin and its expression within the retina changes during various inflammatory and neurodegenerative pathologies. Alarmin is a molecule released from a diseased/damaged cell that initiates an immune response. To discern the cell types in which

IL-33 induction occurs in the hypoxic retina, eyeballs were collected from control (normoxia) and OIR treated mice pups at P15, and the retinas were digested using papain and collagen to obtain single-cell retinal suspensions. The flow cytometric analysis of retinal cell suspensions was carried out to sort out various retinal cells. The CD11b and CD45 antibodies were used to sort out microglia, CD31 antibodies for endothelial cells, CD146 and CD140b for pericytes, and ACSA-1 for astrocytes (Fig. 4a). Neurons were separately sorted out using NeuroFluor NeuO dye (Fig. 4b). To address which retinal cells express IL-33, we performed QRT-PCR analysis on the sorted cells, we observed the expression of IL-33 in microglia, astrocytes, pericytes, and endothelial cells of the retina with no expression of IL-33 in neurons (Fig. 4c). The IL-33 expression was significantly induced in retinal endothelial cells of the OIR group as compared to the normoxia group (Fig. 4c). To obtain additional support on the role of endothelial-specific IL-33 deletion on OIR-induced retinal neovascularization, endothelial specific deletion of IL-33 was executed by giving two intraperitoneal (IP) injections of tamoxifen on P10 and P11 in IL-$33^{flox/flox}$:Cdh5-CreERT2 mice (Fig. 5a). The Cdh5-CreERT2 mice with tamoxifen injections on P10 and P11 were used as control. As shown in Fig. 5b, some residual IL-33 levels were detected in retinal extracts by Western blotting in IL-$33^{flox/flox}$:Cdh5-CreERT2 (IL-$33^{iΔEC}$) mice, apparently due to other cell types in the retina. IL-$33^{iΔEC}$ pups exposed to OIR showed a significant decrease in the number of vessel sprouts and pathological neovascular tufts compared to the control group (Fig. 5c). A similar reduction in retinal neovascularization was observed in IL-$33^{iΔEC}$ mice (Fig. 5d). Notably, the avascular area was similar in IL-$33^{iΔEC}$ mice pups compared to IL-$33^{+/+}$ (Cdh5-CreERT2) control pups (Fig. 5e). In addition, no significant differences were observed in respect to hypoxia-induced retinal neovascularization in EC-specific conditional knockout (IL-$33^{iΔEC}$) mice models compared to IL-33 whole-body knockout mice. Together, these data emphasize the importance of endothelial-specific IL-33 expression on the regulation of pathological angiogenesis, without affecting the normal vascular repair.

**IL-33 activates NFkappaB mediated Jagged1-Notch1 signaling.** Our data demonstrate that IL-33 induced HRMVEC sprouting and tubulogenesis in vitro and retinal EC sprouting and neovascularization in vivo. Therefore, we hypothesized that IL-33 might be regulating Notch signaling. To validate our hypothesis, we looked for the effect of IL-33 on Notch ligands in HRMVECs. IL-33, while having no effect on the steady-state levels of DLL1, DLL3, DLL4, and Jagged2, increased Jagged1 protein levels in HRMVECs (Fig. 6a). We also found that Jagged1 levels were significantly induced in hypoxic retinas compared to normoxia (Fig. 6b). We also observed that hypoxia-induced Jagged1 levels were significantly reduced in IL-$33^{-/-}$ mice compared to IL-$33^{+/+}$ mice (Fig. 6c). In addition, depletion of Jagged1 levels using its siRNA attenuated IL-33-induced sprouting, migration, and tube formation of HRMVECs (Fig. 6d–g).

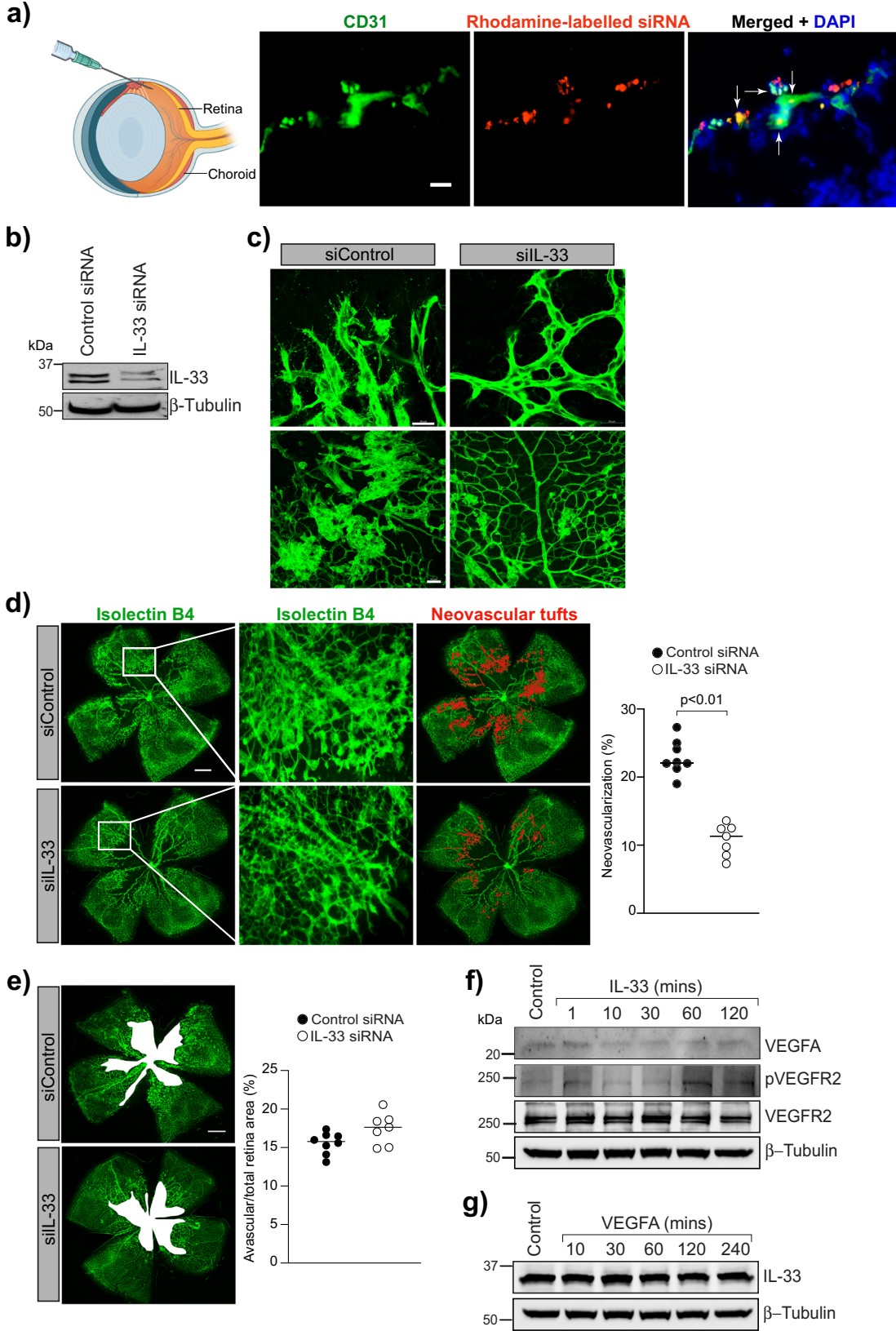

The changes in Jagged1 expression have been implicated in many cancers, and NFkappaB was shown to regulate Jagged1 expression in splenic B cells[34,35]. Therefore, to further understand how IL-33 regulates Jagged1 expression in retinal endothelial cells, we looked for the effect of IL-33 on NFkappaB signaling. IL-33 induces the phosphorylation of IKKalpha/beta, IkappaBalpha,

and NFkappaB in HRMVECs (Fig. 6h) and blockade of NFkappaB activation by QNZ (4-*N*-[2-(4-Phenoxyphenyl) ethyl]-1,2-dihydroquinazoline-4,6-diamine), significantly reduced IL-33-induced Jagged1 expression in HRMVECs (Fig. 6i). We also observed increased activation of NFkappaB in hypoxic retinas (Fig. 6j), and genetic depletion of IL-33 reduced

 **5**

**Fig. 2 IL-33 regulates OIR-induced retinal neovascularization. a** C57BL/6 mice pups were exposed to 75% oxygen (P7–P12), returned to room air and administered intravitreally with 0.5 µl of in-vivofectamine 3-siRNA duplex solution containing 0.6 µg of rhodamine-labeled non targeted siRNA at P12 and P14. At P15, the eyes were enucleated, retinas isolated, fixed, sections made, and stained with anti-CD31 antibodies. The white arrow indicates the colocalization of CD31 with rhodamine labeled siRNA. **b** Everything is same as in **a**, except that mice pup were injected intravitreally with control siRNA (siControl) or IL-33 siRNAs (siIL-33) at P12, and P14. At P17 the retinal tissue extracts were prepared and analyzed for IL-33 levels by Western blotting and normalized to β-tubulin. **c, e** Everything is same as in **b**, except that the retinas were stained with isolectin B4, flat mounts prepared and examined for endothelial tip cell formation (**c**) retinal neovascularization (**d**) and avascular area (**e**). **f, g** Quiescent HRMVECs were treated with and without IL-33 (20 ng/mL) or VEGFA (40 ng/mL) and analyzed for VEGFA, pVEGFR2, VEGFR2 and IL-33 levels using Western blotting and normalized to β-tubulin. The middle column in **d** shows the higher magnification of the area selected. Neovascularization is highlighted in red in the third column of **d**. The bar graphs represent quantitative analysis of neovascularization and avascular area and expressed as Mean ± SD. *$P < 0.05$ vs siControl). Scale bar represents 20 µm in **a**, 20 µm in **c** upper row, 50 µm in **c** lower row, 500 µm in **d** and **e**.

hypoxia-induced NFkappaB activation in retinas (Fig. 6k). These findings suggest the importance of IL-33 induced NFkappaB-Jagged1 signaling in pathological angiogenesis.

The Jagged1 is one of the five ligands for Notch signaling. Therefore, we next looked for the levels of Notch receptors in HRMVECs. IL-33 induces the levels of Notch1 intracellular domain (NICD), with little or no effect on Notch1, Notch2, Notch3, and Notch4 expression (Fig. 7a). The NICD levels were induced in the hypoxic retina, and depletion of IL-33 levels reduces NICD levels in the hypoxic retina (Fig. 7b, c). To understand the role of Notch1 signaling on IL-33-induced angiogenic effects, we looked for the effect of Notch1 signaling on IL-33-induced sprouting, migration, and tube formation in HRMVECs. Knockdown of Notch1 levels with its siRNA molecules significantly reduced IL-33-induced sprouting, migration, and tube formation of HRMVECs (Fig. 7d–g). Together, these data provide compelling evidence that IL-33-induced NFkappaB-Jagged1-Notch1 signaling regulates EC sprouting and tubulogenesis.

**IL-33 enhances Notch1 intracellular domain deubiquitination via its interaction with Numb and BAP1.** Notch signaling not only induces cell division during myogenesis in vertebrates[36,37] but also promotes cellular differentiation and neurogenesis[38,39]. Ubiquitination is a major post-translational modification of Notch, which influences the outcome of Notch signaling. Therefore, we next investigated whether IL-33 influences NICD ubiquitination. We observed that IL-33 treatment results in a time-dependent de-ubiquitination of NICD in HRMVECs (Fig. 8a), and downregulation of Jagged1 with its siRNA enhanced its ubiquitination (Fig. 8b). The E3 ligases and deubiquitinating enzymes regulate the turnover and activity of Notch receptors[40,41]. Therefore, we identified the potential involvement of E3 ligases (FBXW7 and Itch) and deubiquitinating enzymes on IL-33-induced deubiquitylation of NICD. There was no significant effect of IL-33 on FBXW7 and Itch levels in HRMVECs (Fig. 8c). The cell fate determent Numb interacts with E3 ubiquitin ligases to induce the ubiquitination and degradation of NICD[42]. To ascertain the role of IL-33 on Numb and NICD interaction/association, protein from control and IL-33 treated HRMVECs were immunoprecipitated with NICD and immunoblotted for its association with Numb. We observed an increased association of Numb with NICD in IL-33 treated HRMVECs (Fig. 8d). A recent study showed that Numb stabilizes NICD via inducing its interaction with BRCA-associated protein 1 (BAP1, a deubiquitinating enzyme) in cortical neural progenitor cells[43]. As we observed a decreased ubiquitination of NICD in IL-33 treated HRMVECs, we looked for the association of BAP1 with NICD and observed that IL-33 induces the association of NICD with BAP1 and Numb (Fig. 8d). We also observed an increased association of NICD with BAP1 and Numb in OIR retinas compared to normoxia (Fig. 8e). The increased nuclear levels of

NICD, Numb, and BAP1 in IL-33 treated HRMVECs, further validate our observations that IL-33 regulates NICD deubiquitination within the nucleus (Fig. 8f). A schematic diagram depicting the role of IL-33 on Jagged1-Notch1 activation, and pathological retinal neovascularization is presented in Fig. 8g.

**Discussion**

The strategies to treat pathological angiogenesis have shown clinical significance for various proliferative retinopathies[44]. However, the treatment approaches are not selective in only targeting aberrant blood vessels, but also result in inhibition of normal vascular repair, which leads to an increase in ischemic stress. In the present study, we observed that IL-33 levels were induced when new blood vessels were developing in the ischemic retina, and IL-33 induced migration, sprouting, and tube formation of HRMVECs, emphasizing the importance of IL-33 in retinal angiogenesis. The genetic depletion of IL-33 attenuated OIR-induced retinal EC sprouting and neovascularization. These findings indeed suggest a role for IL-33 in pathological retinal neovascularization. It is also interesting to note that IL-33 does not affect normal vascular repair, as its deletion does not affect the avascular area in the hypoxic retina. These observations suggest the therapeutic importance of IL-33 inhibition in various proliferative retinopathies such as diabetic retinopathy, retinopathy of prematurity, and age-related macular degeneration.

IL-33 is considered an alarmin or stress-regulated cytokine, which is mostly expressed by endothelial cells and epithelial cells[45]. Few in-vitro and in vivo studies have shown a role for IL-33 in angiogenesis, notably within the context of inflammation[22,24,46,47]. These studies may not be conclusive as no genetic evidence has been presented supporting the role of IL-33 in angiogenesis. Moreover, its role in post-ischemic neoangiogenesis is unknown. In the present study, we observed that IL-33 levels were induced in endothelial cells of the hypoxic retina and conditional deletion of IL-33 in endothelial cells abrogated OIR-induced retinal EC sprouting and neovascularization. EC-specific deletion of IL-33 in mice pups led to a decrease in ectopic growth of neovascular tufts in the hypoxic retina. These observations may lend further support for the role of IL-33 in retinal neovascularization. We haven't observed any significant differences in retinal neovascularization in EC-specific conditional knockout (IL-33$^{i\Delta EC}$) mice models compared to IL-33 whole-body knockout mice. These observations support our FACS data which clearly shows that retinal endothelial cells were the major contributor for hypoxia-induced IL-33 expression. Vascular endothelial growth factor A (VEGF-A) is an important contributor to ischemia-induced retinal neovascularization[30], and a study has reported that the angiogenic effects of IL-1beta were dependent on VEGFA expression. In this regard, our observations show that IL-33 has no effect on VEGFA expression or VEGFR2 signaling in HRMVECs. Similarly, VEGFA treatment also failed to induce IL-33 expression in HRMVECs. These observations suggest that

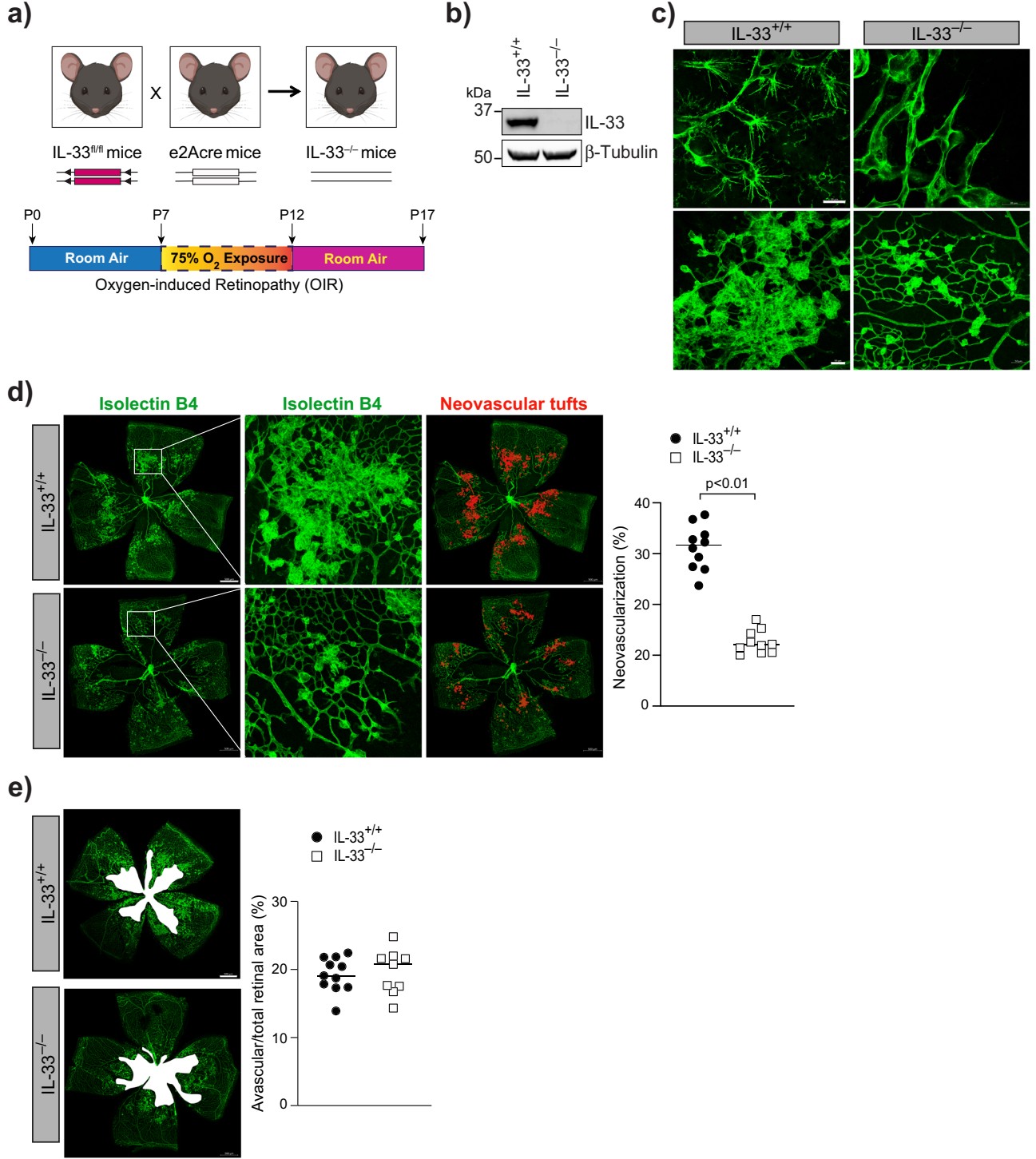

**Fig. 3 Genetic knockdown of IL-33 reduces OIR-induced retinal neovascularization. a** Schematic diagram of breeding strategy for the generation of IL-33 knockout mice. **b** Eyes from IL-33[+/+] and IL-33[−/−] pups were enucleated, retinas isolated, and extracts were analyzed for IL-33 levels by Western blotting and normalized to β-tubulin. **c–e** Everything is same as in **b**, except that retina from IL-33[+/+] and IL-33[−/−] mice pups were stained with isolectin B4, and flat mounts were prepared and examined for endothelial tip cell formation (**c**) retinal neovascularization (**d**), and avascular area (**e**). The middle column in **d** shows the higher magnification of the area selected. Neovascularization is highlighted in red in the third column of **d**. The bar graphs represent quantitative analysis of percentage of neovascularization and avascular area, expressed as Mean ± SD. *$P < 0.05$ vs control siRNA. Scale bar represents 20 μm in **c** upper row, 50 μm in **c** lower row, 500 μm in **d** and **e**.

the angiogenic effects of IL-33 in retinal ECs are independent of VEGF-A-VEGFR2 signaling.

In understanding the mechanisms by which IL-33 influences EC sprouting and vessel formation, we found that IL-33, as well as hypoxia, stimulates Jagged1/Notch1 signaling. Jagged1 is a

Notch ligand that has different biological roles in multiple organs and tissues[48,49]. Overexpression of Jagged1 promotes tumor angiogenesis[17]. However, nothing is known about its role in proliferative retinopathies. To this end, we found that depletion of Jagged1 or Notch1 decreased IL-33-induced sprouting, migration,

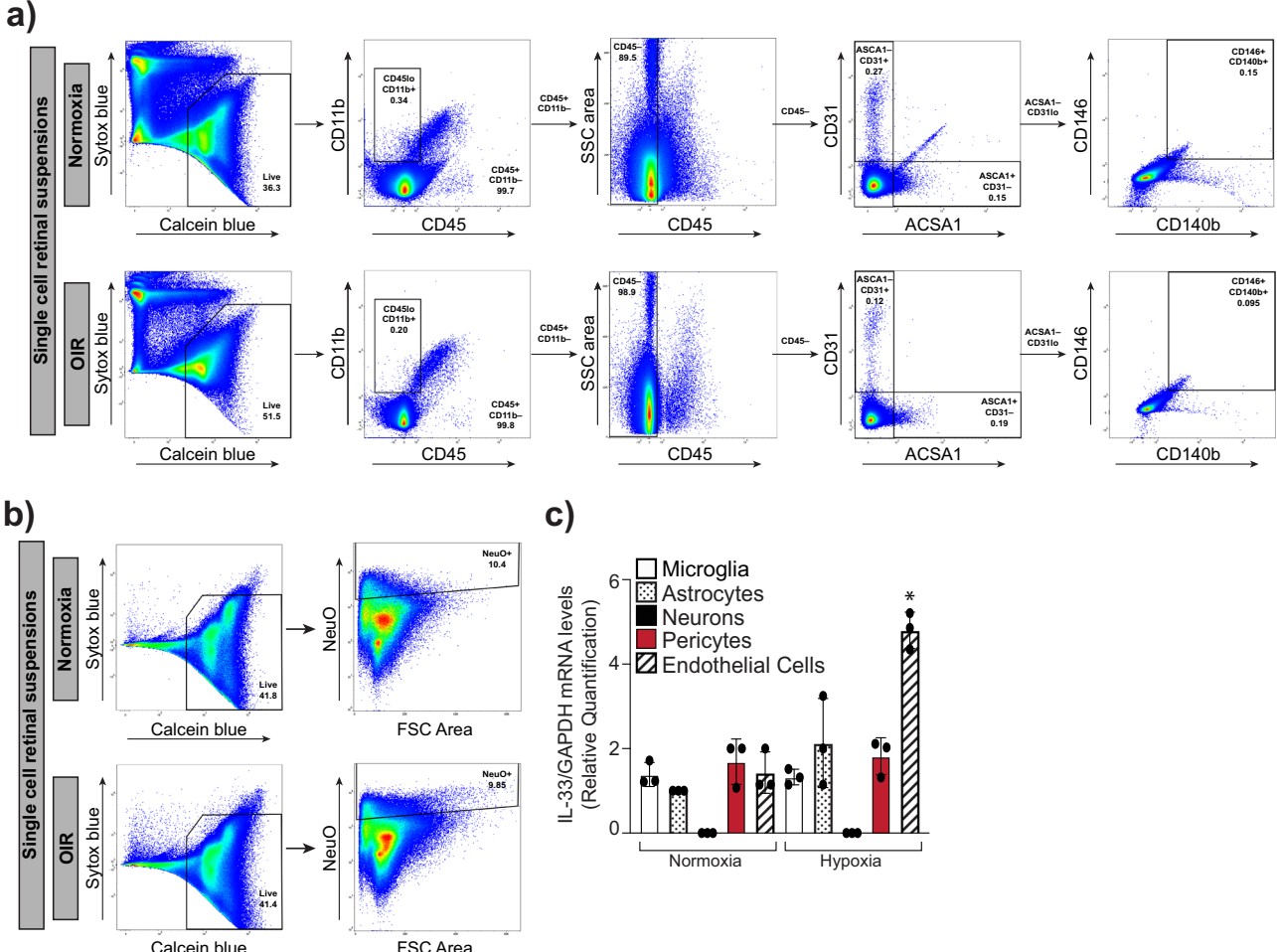

**Fig. 4 Hypoxia-induced expression of IL-33 is mostly confined to retinal endothelial cells.** C57BL/6 mice pups were exposed to OIR and at P15 eyes from pups in normoxia or OIR were enucleated, retinas isolated, minced, digested and single cell retinal suspension were prepared in FACS buffer. The retinal cells were first incubated with indicated dyes and antibodies. After washings, the cells were resuspended into sorting buffer, added SYTOX blue and subjected to FACS analysis. **a** Live cells from both normoxia and P15 were gated as calcein blue positive cells. CD11b positive and CD45 low cells were gated as microglia. CD31 positive and ACSA1 negative cells were gated endothelial cells. ACSA1 positive and CD31 negative cells are gated as astrocytes. CD146 positive and CD140b positive cells are gated as pericytes. **b** Calcein blue positive cells from both normoxia and OIR were gated as live cells and the NeuO+ cells were gated as neurons. **c** Total cellular RNA was isolated from all the sorted cells and analyzed for IL-33 and GAPDH mRNA levels by QRT-PCR. The bar graphs show the quantitative analysis of three independent experiments ($n = 6$ animals/group/experiment), expressed as Mean ± SD. *$P < 0.05$ vs normoxia.

and tube formation of HRMVECs. In addition, genetic deletion of IL-33 decreased Jagged1 expression and Notch1 activation in the hypoxic retina. We also observed that IL-33 induced NFkappaB activation regulates Jagged1 expression in endothelial cells. Our observations are in line with previous findings that IL-33 is a transcriptional regulator of NFkappaB[50], and NFkappaB regulates Jagged1 expression in splenic B cells[35].

Notch1 intracellular domain (NICD) is a short-lived molecule, mostly localized in the nucleus, and many studies have demonstrated that it plays a role in several types of cancers[51,52]. Notch1 or NICD are highly ubiquitinated proteins and the turnover rate of Notch1 signaling depends on NICD ubiquitination/deubiquitination. Interestingly, we observed increased deubiquitination of NICD by IL-33 in retinal endothelial cells, and this phenomenon is dependent on Jagged1 expression. The E3 ubiquitin ligases and deubiquitinating enzymes regulate NICD levels[40,41]. Our in vitro and in vivo findings demonstrate that IL-33 induced association of NICD with Numb and BAP1 (a deubiquitinating enzyme) regulates NICD deubiquitination/stabilization. BAP1 is a ubiquitin carboxy-terminal hydrolase family of deubiquitinating enzymes,

and studies have shown that BAP1 and its orthologs enhance Notch signaling in Drosophila and Zebrafish[53,54]. Although Numb promotes differentiation by antagonizing Notch1 signaling in Drosophila[55], no direct role of Numb on NICD deubiquitination has been reported in higher vertebrates. Moreover, a study has reported that overexpression of Numb did not affect Notch signaling[56]. A recent report supports our observation that Numb binds to NICD and facilitates the recruitment of BAP1, which leads to NICD deubiquitination and stabilization[43].

In summary, our observations identify IL-33 as a potential therapeutic target for pathological retinal angiogenesis, which is a leading cause of blindness in various retinopathies such as ROP, DR, and AMD. We demonstrated that IL-33 regulates sprouting angiogenesis, and vessel anastomosis in retinal endothelial cells, which leads to tufts formation in hypoxic/ischemic retina. The genetic depletion of IL-33 reduces neovascularization in the OIR model with no apparent changes in normal retinal repair. Our findings not only unravel strategies in the amelioration of pathological retinal neovascularization but also have its relevance for translational vascular biology, as growing evidence suggests

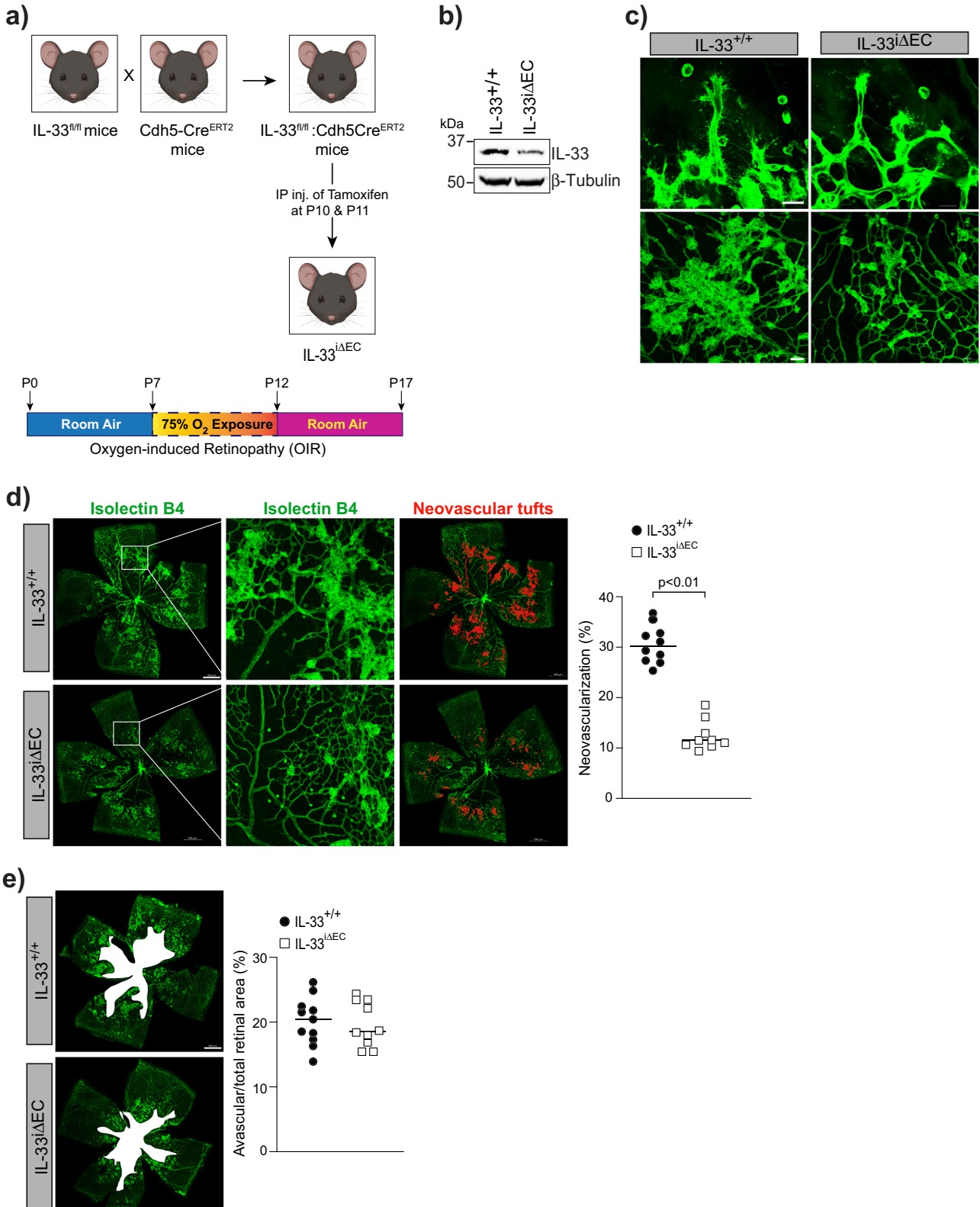

**Fig. 5 Endothelial cells-specific conditional deletion of IL-33 reduces hypoxia-induced pathological angiogenesis. a** Schematic diagram to show the generation of endothelial cells-specific IL-33 knockout mice. **b** Eyes from IL-33$^{+/+}$ and IL-33$^{i\Delta EC}$ pups were enucleated, retinas isolated, and extracts were analyzed for IL-33 levels by Western blotting and normalized to β-tubulin. **c–e** IL-33$^{+/+}$ and IL-33$^{i\Delta EC}$ mice pups were exposed to OIR and at P17 the retinas were isolated, stained with isolectin B4, and flat mounts were examined for endothelial tip cell formation (**c**) retinal neovascularization (**d**), and avascular area (**e**). The middle column in **d** shows the higher magnification of the area selected. Neovascularization is highlighted in red in the third column of **d**. The bar graphs represent quantitative analysis of percentage of neovascularization and avascular area, expressed as Mean ± SD. *$P < 0.05$ vs IL-33$^{+/+}$. Scale bar represents 20 µm in **c** upper row, 50 µm in **c** lower row, 500 µm in **d**, **e**.

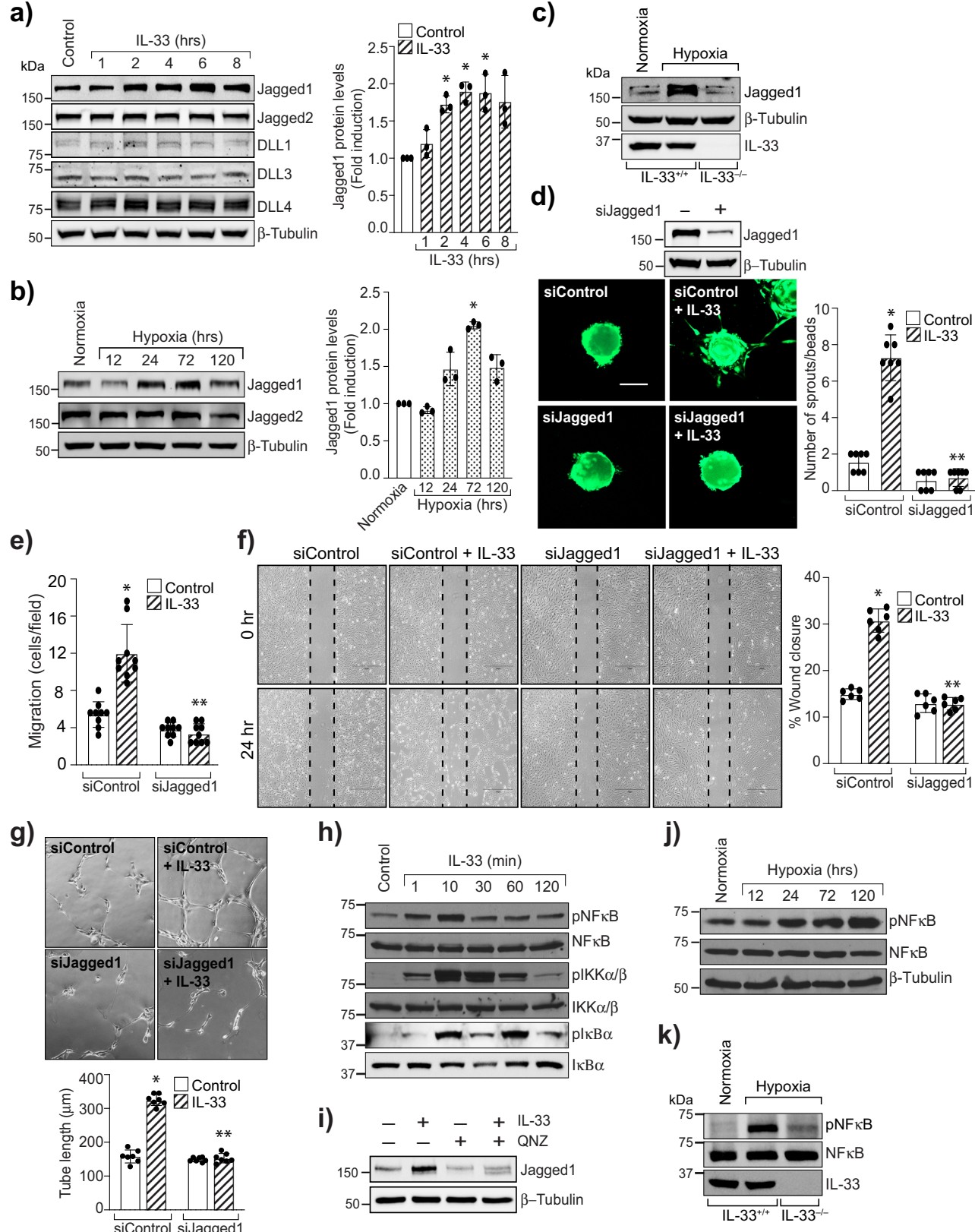

the role for Notch1 signaling in cancer progression and metabolic diseases.

## Methods

**Reagents**. Anti-NOTCH1 (3608, dilution 1:1000), anti-NICD (4147, dilution 1:10007), anti-Notch2 (5732, dilution 1:1000), anti-Notch3 (5276, dilution 1:1000), anti-DLL1 (2588, dilution 1:1000), anti-DLL3 (2483, dilution 1:1000), anti-DLL4 (96406, dilution 1:1000), anti-Jagged1 (2620, dilution 1:1000), anti-Jagged2 (2210, dilution 1:1000), anti-beta-tubulin (2128, dilution 1:1000), anti-Cyclin D1 (55506, dilution 1:1000), and anti-Cyclin D3 (2936, dilution 1:1000) anti-P-VEGFR2 (2478, dilution 1:1000), anti-Numb (2756, dilution 1:1000), anti-Ubiquitin (3936, dilution 1:1000), anti-VEGFR2 (9698, dilution 1:1000) anti-P-NFkappaB (3033, dilution 1:1000), anti-NFkappaB (8242, dilution 1:1000), anti-P-IKKalpha/beta (2697,

**Fig. 6 Jagged1 mediates IL-33-induced angiogenic effects. a** HRMVECs were treated with IL-33 (20 ng/mL) for various time periods, cell extracts prepared and analyzed by Western blotting for Jagged1, Jagged2, DLL1, DLL3, DLL4 levels and normalized to β-tubulin. **b** Retinal tissue extracts from C57BL/6 normoxia and various time periods of hypoxia were analyzed for the indicated proteins by Western blotting and normalized to β-tubulin. **c** At P15, retina tissue extracts from IL-33$^{+/+}$ and IL-33$^{-/-}$ mice pups were analyzed by Western blotting for indicated proteins and normalized to β-tubulin. $n = 6$ animals per group. **d** Upper panel, HRMVECs were transfected with control siRNA (siControl) or Jagged1 siRNA (siJagged1), after 48 h cell extracts were analyzed for Jagged1 levels by Western blotting and normalized to β-tubulin. Lower panel, all the condition are same as in upper panel, except that the cells were analyzed for IL-33 (20 ng/mL)-induced sprouting. **e–g** All the condition are same as in upper panel **d** except HRMVECs were treated with and without IL-33 (20 ng/mL) and subjected to migration (**e**, **f**), and tube formation assay (**g**). **h** Cell extracts treated with various time periods of IL-33 (20 ng/mL) were analyzed by Western blotting for the indicated proteins. **i** Quiescent HRMVECs were first treated with QNZ (10 μM, NFkappaB inhibitor) for 30 min, followed by with or without IL-33 (20 ng/mL) treatment for 2 h. The cell lysates were prepared and analyzed for Jagged1 levels by Western blotting and normalized to β-tubulin. **j** Everything was same as in **b**, except that retinal extracts were analyzed for the indicated proteins by Western blotting. **k** At P15, the retinal tissue extracts from IL-33$^{+/+}$ and IL-33$^{-/-}$ mice pups subjected to normoxia and hypoxia analyzed for indicated proteins by Western blotting. $n = 6$ animals per group. The bar graphs show quantitative analysis of three independent experiments, expressed as Mean ± SD. *$P < 0.05$ vs control siRNA or normoxia, **$p < 0.05$ vs control siRNA + IL-33. Scale bar represents 50 μm in **d**.

dilution 1:1000), anti-IKKalpha (11930, dilution 1:1000), anti-IKKbeta (8943, dilution 1:1000), anti-P-IkappaBalpha (2859, dilution 1:1000), anti-IkappaBalpha (4814, dilution 1:1000), anti-BAP1 (13271, dilution 1:1000), antibodies, were obtained from Cell Signaling Technology (Beverly, MA). Anti-VEGF (sc-57496, dilution, 1:500), anti-MEK (sc-6250, dilution, 1:500), and anti-VEGF (sc-126, dilution, 1:500) antibodies were purchased from Sant Cruz Biotechnology (Dallas, Texas). Growth factor reduced Matrigel (354230) was purchased from BD Biosciences (Bedford, MA). Recombinant anti-NOTCH4 (ab184742, dilution 1:1000) antibody was purchased from Abcam (Cambridge, CA). Human anti-IL-33 (PM033, dilution 1:1000) was purchased from MBL Life Science Company. Recombinant IL-33 (3625-IL-010/CF) protein, Rat anti-IL-33 (MAB3626, dilution 1:1000) and Goat anti-IL-33 antibodies (AF3626, dilution 1:1000) were purchased from R&D systems (Minneapolis, MN). EGM2 medium was purchased from Lonza (Basel, Switzerland). The small interfering RNAs (siRNAs) for Jagged1 [s1176, sense sequence (5′->3′) CCUCAUCCCUGUUACAACA tt, Antisense sequence (5′->3′) UGUUGUAACAGGGAUGAGGgc], NOTCH1 (s9633, sense sequence (5′->3′) GGUGUGCACUGUGAGAUCAtt, Antisense sequence (5′->3′) UGAUCUCACAG UGCACACCct] and scrambled control (4390844) were purchased from Ambion (Carlsbad, CA). Anti-ST2 antibodies (PRS3363, dilution 1:1000), Cytodex 3 microcarrier beads (C3275) and thrombin (T8885) were obtained from SIGMA-ALDRICH (St. Louis, MO). VECTASHIELD Antifade mounting medium with DAPI (H-1500), and without DAPI (H-1700). Bovine fibrinogen (J63276) was purchased from Alfa Aesar (Tewksbury, MA). Prolong Gold antifade reagent (P36984), Cell Tracker Green (C7025), Hoechst 33,342, isolectin B4-594, anti-FBXW7 antibodies (40–1500, dilution 1:1000), anti-Itch antibodies (PA565539, dilution 1:1000), Alexa Fluor 488-conjugated goat anti-rat immunoglobulin G (A11006, dilution 1:250) antibodies were bought from Invitrogen (Carlsbad, CA). Papain suspension (LS003126) and collagenase Type I (LS004194) was obtained from Worthington Biochemical Corporation (Lakewood, NJ). PE anti-mouse CD140b antibody (136005, dilution 1:50), Brilliant Violet 605™ anti-mouse CD45 antibody (103139, dilution 1:25), PerCP/Cyanine5.5 anti-mouse CD146 antibody (134709, dilution 1:25), PE/Cyanine7 anti-mouse CD31 antibody (102523, dilution 1:50), TruStain FcX™ (anti-mouse CD16/32, dilution 1:25) antibody (101319), APC/Fire™ 750 anti-mouse/human CD11b antibody (101261, dilution 1:50) were purchased from Biolegend (San Diego, CA). GLAST (130-123-555, dilution 1:50) antibody was obtained from Miltenyi Biotec (Gladbach, Germany). BD Pharmingen™ BrdU Flow kit (559619) and anti-CD31 (550274, dilution 1:100) were purchased from BDBioscience (San Jose, CA).

**Experimental animals.** C57BL/6 mice were obtained from Charles River Laboratories (Wilmington, MA). IL-33$^{flox/flox}$ mice (stock number 030619), and E2a-Cre mice (stock number 003724) were obtained from Jackson Laboratory (Bar Harbor, ME). The mice were bred and housed in a 12-h light/12-h dark cycle environment and fed ad libitum food and water. They were housed in Wayne State University DLAR animal facility, Detroit, MI. Both female and male mice pups (of age P12 to P17) were used for this study. All the animal experiments were approved by the Animal Care and Use Committee of Wayne State University, Detroit, MI.

*Generation of IL-33 knockout mice.* To generate IL-33 knockout mice, we interbred IL-33$^{flox/flox}$ mice[57] with E2a-Cre mice. E2a-Cre recombinase leads to germ line deletion of IL-33 in mice[58].

*Generation of IL-33 conditional knockout mice.* To knockout IL-33 postnatal in the retinal vasculature, IL-33$^{flox/flox}$ mice were interbred with transgenic mice expressing endothelial Cdh5 promoter controlling the recombinase (CreERT2) and induced by the tamoxifen injections[59]. IL-33$^{flox/-}$:Cdh5-CreERT2 mice were interbred with IL-33$^{flox/flox}$ to generate litters containing IL 33$^{flox/flox}$:Cdh5-CreERT2 and control (IL-33$^{flox/flox}$) littermates. For stimulating Cre activity and IL-33 gene deletion, two tamoxifen injections (Sigma-Aldrich, T5648; 2 mg/ml; dissolved in 1:9 ratio of

ethanol and corn oil) of 50 μl was given IP to pups at P10 and P11 to make endothelial-specific deletion of IL-33 (IL-33$^{iΔEC}$).

**Cell culture.** The human retinal microvascular endothelial cells (HRMVECs, # ACBRI 181) were obtained from Cell Systems (Kirkland, WA). EGM2 medium supplemented with gentamycin (10 μg/mL), and amphotericin B (0.25 μg/mL) was used for growing HRMVECs. Cell cultures were maintained at 37 °C in incubator with 95% air and 5% CO$_2$. The HRMVECs were synchronized by incubating them in serum-free EBM2 medium for 24 h and then used for experiments.

**Cell migration.** HRMVECs migration assay was carried out using a modified Boyden chamber method[60]. Quiesced HRMVECs were suspended in EBM2 medium and were plated on growth factor reduced Matrigel-coated-8-μm cell culture inserts at a concentration of $5 \times 10^4$ cells per insert. Vehicle or IL-33 were added to the lower chamber at the specified concentrations. To verify the effect of indicated siRNA on IL-33-induced HMVECs migration, cells were first transfected with control or test siRNA, growth-arrested overnight in serum free media and then subjected to migration at 37 °C. The non-migrated cells present on the upper side of the membrane were removed using sterile cotton swabs, and the migrated cells on the lower surface of the membrane were fixed for 15 min in methanol. Thereafter, membrane with cells were stained with DAPI (Vector Laboratories) and the slides were observed under EVOSM5000 Imaging System with 10× magnification (Thermo Fisher Scientific, Waltham, MA). The migrated cells were counted in 7–10 randomly selected fields and plotted as migrated cells per field.

**Wound healing assay.** HRMVECs were plated in six-well culture dishes, left grown to full confluency and growth-arrested for 24 h at 37 °C. After a period of growth arrest, a sterile plastic micropipette tip was used to simulate a wound by making a straight-edged and cell-free zone in each well. The cells were washed with EBM2 media and then treated with and without IL-33 (20 ng/mL) for 24 h in EBM2 media containing 5 mM hydroxyurea. The migration of cells was observed under EVOS M5000 Imaging System at 10× magnification (Thermo Fischer Scientific, Waltham, MA) and NIH ImageJ version 1.43 software was used to analyze the wound closure. The percentage of wound closure was measured to assess the cell migration of HRMVECs [total wound area (0 h) − wound area (24 h)/total wound area (0 h) x 100].

**DNA synthesis.** The [$^3$H]-thymidine incorporation assay was used to measure DNA synthesis. Briefly, the HRMVECs were first quiesced, trypsinized, and plated on 24-well plate at a concentration of $2 \times 10^5$ cells per mL of the medium. IL-33 (20 ng/mL) or vehicle were added to the medium and the cells were incubated for 6 h. After 6 h, the HRMVECs were pulse-labeled with 1 μCi/ml of [$^3$H]-thymidine for 24 h. The cells were then washed with cold PBS and collected in 3 ml of cold EDTA (5 mM). Three milliliters of 20% (w/v) cold trichloroacetic acid (TCA) was then added to cells and vortexed vigorously to lyse the cells. The cells were then incubated on ice for 30 min and then filtered through a GF/F glass microfiber filter. The filter was then washed with cold TCA (5%) and then with cold ethanol. The filter was dried, placed in a vial containing liquid scintillation fluid and the radioactivity was measured using a liquid scintillation counter (Beckman LS3801). The counts/min/well were measured to assess DNA synthesis.

**BrdU cell proliferation assay.** The BrdU measurement was conducted according to the standard protocol of manufacturer (BD biosciences). HRMVECs were cultured in the 100-mm dishes with density $2 \times 10^6$ cells/mL, and then growth arrested. Following the growth arrested period, cells were treated with IL-33 at a concentration of 20 ng/mL and incubated for 24 h. BrdU (10 μl/mL, 1 mM) was added and incubated for 4 h at 37 °C. The cells were then fixed and permeabilized with BD cytofix/cytoperm buffer and kept at room temperature (RT) for 30 min. Thoroughly after removal of the fix solution, the cells were washed with 1× BD

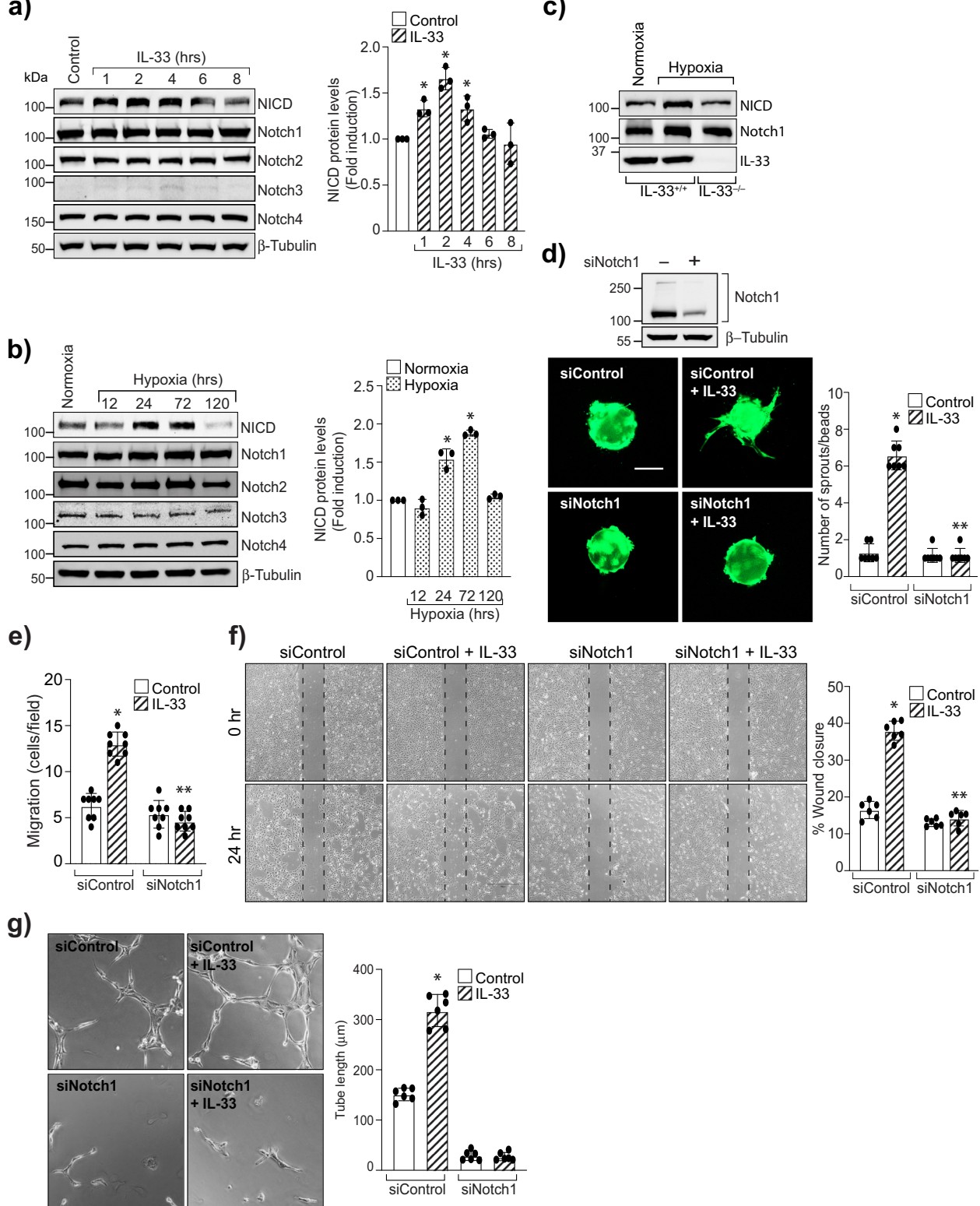

**Fig. 7 Notch1 activation mediates IL-33-induced angiogenic effects. a** Quiesced HRMVECs were treated with and without IL-33 (20 ng/mL) for various time periods and analyzed by Western blotting for Notch1 intracellular domain (NICD), Notch1, Notch2, Notch3, Notch4 levels and normalized to β-tubulin. **b** Retinal extracts of C57BL/6 mice pups from normoxia and various time periods of hypoxia were analyzed for the indicated proteins by Western blotting and normalized to β-tubulin. **c** At P15, retinal extracts from IL-33$^{+/+}$ and IL-33$^{-/-}$ mice pups were analyzed by Western blotting for indicated proteins. $n = 6$ animals per group. **d** Upper panel, HRMVECs were transfected with control (siControl), or Notch1 siRNA (siNotch1), cell extracts were analyzed for Notch1 level by Western blotting and normalized to β-tubulin. Lower panel, all the condition are same as in upper panel, except that the cells were analyzed for IL-33 (20 ng/mL)-induced sprouting. **e–g** All the condition are same as in upper **d**, except those cells were subjected to IL-33 (20 ng/mL)-induced migration (**e**, **f**), and tube formation (**g**). The bar graphs show quantitative analysis of three independent experiments, expressed as Mean ± SD. *$P < 0.05$ vs control siRNA or normoxia, **$p < 0.05$ vs control siRNA + IL-33. Scale bar represents 50 μm in **d**.

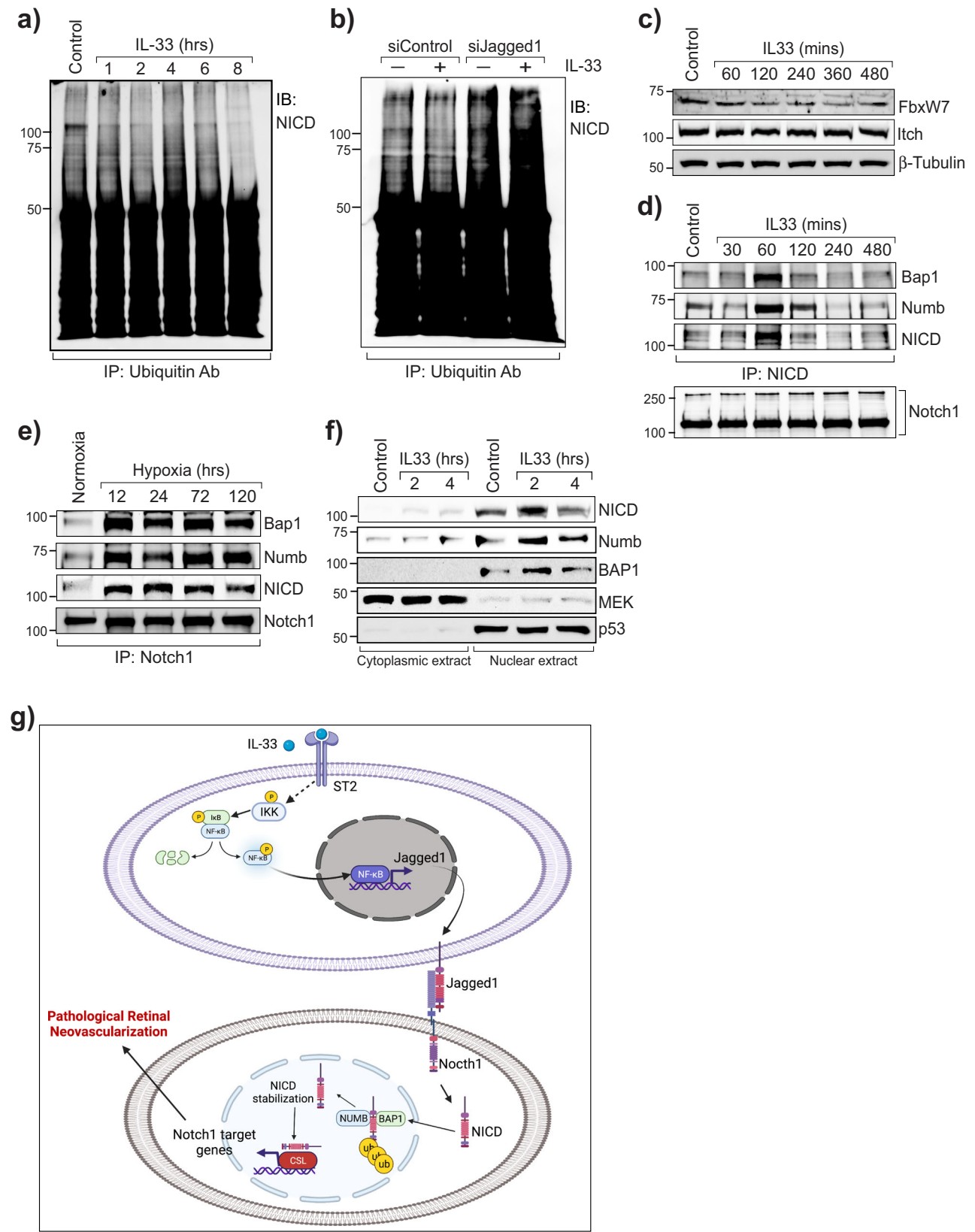

perm/wash buffer. The cells were permeabilized again with BD cytoperm permeabilization buffer plus and incubated for 10 min on ice. After washing, the cells were then again fixed with BD cytofix/cytoperm buffer for 5 min at RT. The cells were then incubated with DNase solution for 1 h at 37 °C. After washing, 50 microliter of working solution of FITC conjugated anti-BrdU antibody was added, and the cells were incubated for 20 min at RT. The cells were washed and then 20 µl

of the 7-AAD solution was added. Lastly, the HRMVECs were suspended in 1 mL of staining buffer. The samples were then analyzed using BD LSRII flow cytometer.

*Sprouting assay*. HRMVECs three-dimensional sprouting assay was carried out using Cytodex 3 microcarrier beads[61]. The HRMVECs were transfected with the control and indicated siRNAs, trypsinized, and incubated with Cytodex 3

**Fig. 8 IL-33 enhances Jagged1 mediated Notch1 de-ubiquitination and stabilization. a** Quiescent HRMVECs were treated with or without IL-33 (20 ng/mL), and cell extracts were immunoprecipitated (IP) with anti-Ubiquitin antibodies, and immunoblotted (IB) with Notch1 intracellular domain (NICD) antibodies. **b** HRMVECs were transfected with control (siControl), or Jagged1 siRNA (siJagged1), quiesced, treated with and without IL-33 (20 ng/mL) for 1 h and cell extracts were immunoprecipitated with anti-Ubiquitin antibody and immunoblotted for NICD levels. **c** Everything is same as in **a**, except that the cell extracts were analyzed by Western Blotting for the indicated proteins. **d** Upper panel, all the conditions were same as in **a**, except that cell lysates were immunoprecipitated (IP) with anti-NICD antibodies, and immunoblotted (IB) with BAP1, Numb and NICD antibodies. Lower panel, the cell extracts from upper panel were analyzed by Western blotting for Notch1 levels. **e** Normoxic and hypoxic retinal extracts from C57BL/6 mice pups were immunoprecipitated (IP) with anti-Notch1 antibodies, and immunoblotted (IB) for the indicated proteins. **f** HRMVECs were quiesced, treated with IL-33 (20 ng/mL) for 2 or 4 h, cytoplasmic and nuclear extracts were prepared and analyzed for NICD, Numb and BAP1 by Western blotting and normalized with MEK1 (cytoplasmic marker) and p53 (Nuclear marker). **g** Schematic diagram depicting the role of IL-33 on Jagged1-Notch1 activation, and pathological retinal neovascularization.

microcarrier beads at 37 °C overnight for the attachment of cells to the beads. The beads were then embedded in the 3D bed of fibrin gels and fibroblasts were seeded onto the topical surface of the fibrin gels. Cell sprouting was observed on day 3 using Zeiss LSM 800 microscope and the images were captured using image analysis software Zen.

*Tube formation.* HRMVECs tube formation assay was performed using growth factor reduced matrigel[62]. Full confluent HRMVECs were growth-arrested, platted in 24-well culture dishes, already coated with growth factor-reduced Matrigel. In order to verify the effect of siRNAs, cells were transfected with control and specific siRNAs, quiesced, and then subjected to tube formation assay. The cells were incubated with vehicle or IL-33 (20 ng/mL) at 37 °C for 6 h. After the incubation, HRMVECs tube formation was examined under EVOS M5000 Imaging System at 10X magnification (ThermoFisher Scientific, Waltham, MA). The NIH ImageJ version 1.43 software was used to calculate HRMVECs tube length and expressed in micrometers.

*Western blotting.* To perform western blotting, an equal amount of protein from cells or tissue extracts were electrophoretically resolved on SDS-PAGE gels. The proteins resolved on the gels were then transferred to a nitrocellulose membrane. The membranes were then blocked in either 5% (w/v) nonfat dry milk or bovine serum albumin. After blocking, the nitrocellulose membranes with transferred proteins were probed with the suitable primary and secondary antibodies. The antigen–antibody complexes on the membranes were detected using an enhanced chemiluminescent detection (Supersignal West Pico Plus, Thermofisher Scientific).

*Transfections.* The cells were transfected with control or indicated siRNA molecules using lipofectamine 3000 transfection reagent at a final concentration of 100 nM. Following transfections, HRMVECs were growth-arrested in EBM2 (serum-free) medium and used for further experiments.

*Intravitreal Injections.* Mice pups were intravitreally administered with 0.5 microliter of invivofectamine 3-siRNA duplex solution containing 0.6 µg of control or IL-33 siRNA at P12 and P14 using a 35G needle.

*Immunofluorescence staining.* Following hyperoxia exposure, mice pups were administered with rhodamine labeled-nontargeted siRNA at P12 and P14. At P15, the pups were euthanized, eyes enucleated, and embedded in optimal cutting temperature compound. 10-µm cryosections were made from the retina central part. To know the location of rhodamine labeled-nontargeted siRNA in the retina, the sections were probed with rat anti-mouse CD31(1:100) primary antibodies and Alexa Fluor 488-conjugated secondary antibodies. These sections were then examined under Zeiss LSM800 confocal microscope and images were captured using image analysis software Zen (Carl Zeiss Imaging Solutions GmbH).

*Oxygen-induced retinopathy (OIR).* At P7, mice pups along with dams were kept in BioSpherix chamber for five days and exposed to 75% oxygen (P7–P12), and then returned to room air[63]. The mice littermates of the same age continued at room air (21% oxygen), were considered as controls. At P17, the pups were euthanized, their eyes enucleated and fixed. The retinas were isolated and stained with rhodamine labeled isolectin B4. After staining, the retinas were flat mounted and observed under a Zeiss LSM 800 confocal microscope and retinal neovascularization was quantified using Nikon NIS-Elements advanced research software[60]. Retinal neovascularization was highlighted in red, and the retinal neovascularization was calculated as fluorescence intensity of the highlighted area/total fluorescence intensity of the retina × 100. The percent avascular area was defined as total avascular area/total retina area × 100.

*Primary retinal cell isolation.* The mice were first anesthetized and then perfused with PBS before eye enucleation. The retinas were isolated, minced, and digested in 15 µg/mL DNase and 15 IU/mL papain solution for 30 min at 37 °C. Through gentle pipetting the tissue was dissociated and then passed through a 40 µm cell strainer. The flow through were mixed with fetal bovine serum (FBS) and washed

with DMEM containing 10% FBS. The trapped tissue in 40 µm cell strainer was digested using collagenase type I (1 mg/mL) for 30 min at 37 °C to obtain retinal endothelial cells. Cells obtained after collagenase and papain-DNase digestion were pooled and counted.

*Retinal cell sorting by flow cytometry.* Suspensions of primary retinal cells from nine mice each from normoxia and hypoxia groups were pooled and incubated with Calcein Blue and SYTOX Blue (Thermo Fisher Scientific, Waltham, MA) for 20 min on ice. After which FACS buffer was used to wash cells and incubated for 10 min with Mouse Fc block (anti-CD16/CD32, clone 2.4G2) to minimize non-specific binding to Fc receptors. Thereafter, cells were incubated with the following antibodies: APC/Fire750-conjugated CD11b, BV605-conjugated CD45, PE-Cy7-conjugated CD31, PerCP-Cy5.5-conjugated CD146, PE-conjugated CD140b (Bio-Legend, San Diego, CA), and APC-conjugated ACSA-1 (Miltenyi Biotec, San Diego, CA). Separately, neurons were stained with NeuroFluor NeuO (StemCell Technologies, Cambridge, MA). The gates for sorting were set based on fluorescence-minus-one controls. Cell sorting was performed on an SY3200 cell sorter (Sony Biotechnology, San Jose, CA). The sorted cells were analyzed by QRT-PCR for IL-33 expression.

*Quantitative real-time-PCR (QRT-PCR).* The total cellular RNA was isolated from retina using Trizol. High-Capacity cDNA reverse transcription kit was used to prepare cDNA following manufacture's protocol (Applied Biosystems, Foster City, CA). The TaqMan Gene Expression Assays for mouse IL-33 (Mm00505403), and mouse GAPDH (Mm99999915_g1) were used for PCR amplification. The PCR amplification was carried out on 7300 Real-Time PCR Systems (Applied Biosystems) using the manufacturer's cycling parameters. The PCR amplification run, and Ct values were gathered using 7300 real-time SDS version 1.4 program.

*Statistics and reproducibility.* All the experiments were repeated at a minimum of three times, and data in the graphs are presented as mean ± SD. Two tailed *t* tests were used to calculate differences between two groups. For experiments having more than two groups, comparisons were assessed using one-way ANOVA with Bonferroni's correction. GraphPad Prism 9 (Prism) was used for statistical analyses. All $P < 0.05$ values were statistically significant.

**Reporting summary**. Further information on research design is available in the Nature Research Reporting Summary linked to this article.

## Data availability

The supplementary figures 3–10 contain all the uncropped Western blot images presented. The statistical source data for all the graphs presented in figures are available as Supplementary Data 1 (Excel file), and the remaining datasets are available from the corresponding author on request.

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

## Acknowledgements

The present research work was supported by National Institutes of Health grants (EY029709 to N.K.S. and EY014856 to G.N.R.). Supported by a Research to Prevent Blindness unrestricted grant to Kresge Eye Institute, and by P30EY04068 (L.D.H.) at Wayne State University. The Microscopy, Imaging and Cytometry Resources Core is supported in part by NIH Center grant P30 CA22453 to the Karmanos Cancer Institute

and R50 CA251068-01 to Kamiar Moin, Wayne State University. The Schematic diagram of the present research is created with BioRender.com.

## Author contributions

D.S. performed cell migration, sprouting, proliferation, tube formation, OIR, and Western blot analysis and wrote the manuscript. S.B. performed OIR and Western blot analysis. G.K. performed Western blot analysis. E.C.V.B., performed FACS analysis. G.N.R. provided valuable suggestions and editited the manuscript. N.K.S. performed cell migration, sprouting, DNA synthesis, Western blot analysis, immunofluorescence staining, OIR, designed the project, supervised the study, interpreted the data and wrote the manuscript.

## Competing interests

The authors declare no competing interests.
