## [Peer Review File · Communications Biology]

Reviewers' comments:

Reviewer #1 (Remarks to the Author):

In "IL-33 enhances Jagged1 mediated NOTCH1 intracellular domain (NICD) deubiquitination and pathological angiogenesis in a murine model of proliferative retinopathy" the authors present evidence of a novel induction of the Jagged/Notch axis with high relevance for neoangiogenic mechanisms and significance for the field of proliferative retinopathies. The manuscript is well written scientifically, with clear and carefully presented data, in line with the discussion and conclusions presented. Nevertheless, so aspects should be considered for revision.

Major review

1. In introduction, last sentence of paragraph 2, the authors claim that "Jagged1 has never been studied in pathological angiogenesis." The statement is not truthful as the authors also disclose (discussion, paragraph 3, sentence 3) the role of Jagged1 is well characterized in tumor neoangiogenesis, particularly intussusceptive neoangiogenesis. Please rewrite the paragraph to a better structured statement, that does not require discussion.

2. A similar comment is claimed in discussion (paragraph 3, sentence 4) where the authors claim Jagged1 function/role is unknown in the proliferative retinopathies. Again, the statement is an exaggeration, since the role of Notch signaling has been established in retinal angiomatose proliferation (nowadays, macular neovascularization type 3) as well as retinal neoangiogenic tumors (retinoblastoma), where intussusceptive neoangiogenesis of retinal vasculature. The sentence should be deleted, or at least very carefully discussed.

3. In discussion, paragraph 2, sentence 5, the authors make reference to "decreased anastomoses in hypoxic retinas." The authors do not present evidence of reduced anastomoses, rather only vascular tufts. Please delete the mention of anastomosis.

4. In results, subsection 1, paragraph 1, sentences 3 and 6, make reference to data not shown on expression of several cytokines and interleukins, yet only IL-33 is presented. The claims is of high relevance for the manuscript and must be disclosed and/or cited to previous references. If the authors are considering other future publication based on the disclosed data set, and are not ready to disclose such information, the paragraph must be fully re-written to refer only and exclusively to IL-33 expression and justified.

5. On figures and multiple instances, no SD/SEM is presented for controls in graphs. This would directly invalidate the statistical methods applied to the data (a zero value on SD creates a mathematical impossibility on T-tests). Graphs must be revised.

Minor review

1. Although scientifically well written, multiple issues with english language were noted. some examples below, yet the text would benefit a general english revision.

1a. the introduction is at times written in the present, should be in the past participle (eg. "is shown" should be "have been shown")

1b. When referring to "vascular plexus" is a singular; "vascular plexi" for plural (introduction, paragraph 1)

1c. many acronyms change formate (eg. Jagged-1 vs Jagged1; same doe DLLs), please uniformize.

1d. A more careful disambiguation of acronyms on first entry should be revised.

1e. "neoangiogenesis," "In vitro/vivo" should not be hyphenated.

1f. On nomenclature: Vegfa, gene; VEGFA, transcript; VEGF-A, protein.

1g. multiple articles (direct and indirect; eg. The), prepositions, and third-person verbs, are misused/conjugated.

2. On figures, most graphs referring to OIR are more commonly presented as post-natal days (eg. P17) since it relates to the model ages as well characterized in the literature. Review the graph legends to mirror the standard format.

3. also on graphs, on some figures it is difficult to reader to distinguish between protein/transcript quantification, could be added to y-axis legends.

4. In methods, statistics: some graphs are presented as SD others as SEM, please re-write the subsection. Suggestion "as mentioned in figure legends"

Reviewer #2 (Remarks to the Author):

Comments to authors

General comments

Authors should keep in mind the basic principle of the OIR model. For example, what is defined as pathology in this model? What is defined as neovascularization? Where do these different responses occur? What is the biggest challenge in the management of ROP. Answers to this question have significant implications for conducting and interpreting experiments in the OIR model. I am afraid these aspects were not clearly elaborated in the current study, and yet this has interpretation implications.

Specific comments

1. I think authors meant to write that DLL4 is the most characterized ligand for Notch1 activation and not DLL4 is the most characterized receptor for Notch1 activation as currently written in the introduction.

2. In the introduction, authors claim that Jagged 1 has never been studied in angiogenesis, however, I would like to draw the authors attention to publications where jagged 1 was studied in angiogenesis. <https://pubmed.ncbi.nlm.nih.gov/26213336/>, <https://pubmed.ncbi.nlm.nih.gov/28445154/>, and <https://www.mdpi.com/1422-0067/21/18/6477>, among others. As such, this statement should be revised, and the relevant literature cited.

3. In the methods on page 18, optimal cutting temperature compound is not a fixative, but rather an embedding medium. Please correct this statement.

4. On page 18, description of the OIR model, a known region of the retina does not develop blood vessels due to exposure to hyperoxia, and upon return to normoxia, neovascularization occurs in a defined region of the tissue. As such, I recommend the analysis of the vascularization and neovascularization be limited to these areas of the tissue, and representative high resolution zoomed in images of these areas be presented in the corresponding results.

5. Authors should provide conclusive evidence supporting the idea that the effects of IL-33 siRNA were because of its effect on the retinal vascular cells and not any other cell type in the retina.

6. Authors write on page 6 of the methods that "OIR model is a well-characterized VEGFA-dependent model and suppression of VEGFA expression or activity inhibits neovascularization in several diseases", but, in the opening paragraph of the results authors write "Current anti-angiogenic therapies are focused mostly on VEGFA neutralization or blockade of it signaling. However, several patients suffering from age-related macular degeneration or cancer do not respond to anti-VEGFA therapies". One wonders what the point of this study is if it anyway targets

VEGF which is already shown to be of limited efficacy?

Reviewer #3 (Remarks to the Author):

This is a wonderful study that examined the importance of IL-33 in hypoxia-induced retinopathy. IL-33 is not only necessary for emergence of neovascularization in the OIR model, but also appears to act independently of VEGF. This offers good prospects for alternatives to therapy in PDR patients. The mechanism of the notch pathway involved is also examined in detail. The study is very clearly described and the methods used are relevant and well chosen.

Some suggestions for improvement are given below.

Abstract: the abstract is unclear at some points because of a vague description of methods used. It would be more appealing to the reader to better understand in which models in vivo or in vitro the findings were made.

Introduction: It will be interesting to explain why IL-33 was selected out of all other cytokines and interleukins.

Results:

-The authors state in the 1st paragraph "The retinas from various periods of relative hypoxia were analyzed for the expression of cytokines and interleukins." What other cytokines and interleukins were investigated and where is the data? If only IL-33 was investigated, this sentence should be adjusted.

-2nd paragraph: authors can describe more about Fig1Band C, because there seems to be an optimum at 72 hours, whereas IL-33 levels are back to normal at 120h.

-It is unclear what the term "angiogenic events" means, instead use "angiogenic effects" or "angiogenic properties".

-Fig2A shows possible colocalization of siRNA against IL-33 and ECs (CD31). However, this is difficult to see from a 2D projection. It cannot be excluded that a supporting cell type (e.g pericyte) contains the siRNA. This can be better visualized by showing also the z-axis. Furthermore, it is also clear that other cells took up the siRNA. Please mention this in the results. After all, what is the importance of EC-specific knockdown? Isn't IL-33 secreted and then it should not matter what cells express it?

-The 60% reduction in protein levels should also be presented in a bar graph or do not mention "60%"

-"pathological aberrant" is a pleonasm

-Explain what is an "alarmin"

-Page7 "This fact can be very important": change fact to finding.

-Fig4 shows the rationale for looking for localization of IL-33 siRNA in ECs, since induction of IL-33 is only happening in ECs. Thus, for a logical storyline, Fig4 would better fit prior to figure 2A.

-Did the authors see any differences in germ line deletion versus conditional knockout of IL-33 in the OIR experiments? Please describe in Results and address this in the Discussion.

-The title of Fig7 "Jagged1 dependent Notch1 activation mediates IL-33-induced angiogenic events" is scientifically incorrect, as it has not been shown that knockdown of jagged1 leads to less notch activation.

We are thankful for your kind decision letter and the Reviewers' comments on our manuscript entitled "IL-33 enhances Jagged1 mediated NOTCH1 intracellular domain (NICD) deubiquitination and pathological angiogenesis in proliferative retinopathy." Based on Reviewer #1, #2 and #3 comments or suggestions, we have extensively revised the manuscript and incorporated the changes in the revised manuscript. All the changes in the manuscript are marked as red.

Answers to Reviewer #1:

Major Review

1. In introduction, last sentence of paragraph 2, the authors claim that "Jagged1 has never been studied in pathological angiogenesis." The statement is not truthful as the authors also disclose (discussion, paragraph 3, sentence 3) the role of Jagged1 is well characterized in tumor neoangiogenesis, particularly intussusceptive neoangiogenesis. Please rewrite the paragraph to a better structured statement, that does not require discussion.

Answer: *In response to Reviewer #1's suggestions, we have now changed it in the revised manuscript (please refer to page4, lines 97-99).*

2. A similar comment is claimed in discussion (paragraph 3, sentence 4) where the authors claim Jagged1 function/role is unknown in the proliferative retinopathies. Again, the statement is an exaggeration, since the role of Notch signaling has been established in retinal angiomas proliferation (nowadays, macular neovascularization type 3) as well as retinal neoangiogenic tumors (retinoblastoma), where intussusceptive neoangiogenesis of retinal vasculature. The sentence should be deleted, or at least very carefully discussed.

Answer: *In response to Reviewer #1's suggestions, we have now deleted it in the revised manuscript.*

3. In discussion, paragraph 2, sentence 5, the authors make reference to "decreased anastomoses in hypoxic retinas." The authors do not present evidence of reduced anastomoses, rather only vascular tufts. Please delete the mention of anastomosis.

Answer: *We deleted it in the revised manuscript.*

4. In results, subsection 1, paragraph 1, sentences 3 and 6, make reference to data not shown on expression of several cytokines and interleukins, yet only IL-33 is presented. The claims is of high relevance for the manuscript and must be disclosed and/or cited to previous references. If the authors are considering other future publication based on the disclosed data set, and are not ready to disclose such information, the paragraph must be fully re-written to refer only and exclusively to IL-33 expression and justified.

Answer: *In response to Reviewer #1's suggestions, we have now re-written the section in the revised manuscript to include a justification (please refer to page 5, lines 133-139).*

5. On figures and multiple instances, no SD/SEM is presented for controls in graphs. This would directly invalidate the statistical methods applied to the data (a zero value on SD creates a mathematical impossibility on T-tests). Graphs must be revised.

Answer: In relative quantification or fold induction graphs, we usually keep control

as a calibrator and give it a value of 1. In each set of experiments, the control is kept as 1 and then fold induction is calculated. For example, when we perform three experiments, then we have three controls, which have a value of 1. Therefore, the SD/SEM of controls comes as zero. This is a standard way of presentation for fold induction graphs, and T-tests evaluation.

Minor review

1. Although scientifically well written, multiple issues with English language were noted. Some examples below, yet the text would benefit a general English revision.

Answer: We revised the manuscript.

2. 1a. The introduction is at times written in the present, should be in the past participle (eg. "is shown" should be "have been shown")
Answer: changed.
3. 1b. When referring to "vascular plexus" is a singular; "vascular plexi" for plural (introduction, paragraph 1)
Answer: modified.
4. 1c. Many acronyms change format (eg. Jagged-1 vs Jagged1; same for DLLs), please uniformize.
Answer: incorporated.
5. 1d. A more careful disambiguation of acronyms on first entry should be revised.
Answer: revised.
6. 1e. "neoangiogenesis," "In vitro/vivo" should not be hyphenated.
Answer: modified.
7. 1f. On nomenclature: Vegfa, gene; VEGFA, transcript; VEGF-A, protein.
Answer: incorporated.
8. 1g. Multiple articles (direct and indirect; eg. The), prepositions, and third-person verbs, are misused/conjugated.
Answer: modified.
9. On figures, most graphs referring to OIR are more commonly presented as post-natal days (eg. P17) since it relates to the model ages as well characterized in the literature. Review the graph legends to mirror the standard format.
Answer: modified.
10. 3. Also on graphs, on some figures it is difficult for the reader to distinguish between protein/transcript quantification, could be added to y-axis legends.
Answer: modified.
11. In methods, statistics: some graphs are presented as SD others as SEM, please re-write the subsection. Suggestion "as mentioned in figure legends"
Answer: modified (please refer to page 21, lines 605-607).

Answers to Reviewer #2:

1. I think authors meant to write that DLL4 is the most characterized ligand for Notch1 activation and not DLL4 is the most characterized receptor for Notch1 activation as currently written in the introduction.

Answer: *We are thankful to the reviewer for his/her suggestions, and we have corrected it in the revised manuscript (please refer to page 3, line 91).*

2. In the introduction, authors claim that Jagged 1 has never been studied in angiogenesis, however, I would like to draw the authors attention to publications where jagged 1 was studied in angiogenesis. <https://pubmed.ncbi.nlm.nih.gov/28445154/>, and <https://www.mdpi.com/1422-0067/21/18/6477>, among others. As such, this statement should be revised, and the relevant literature cited.

Answer: *We have now modified it in the revised manuscript (kindly refer to page 4, lines 97-99).*

3. In the methods on page 18, optimal cutting temperature compound is not a fixative, but rather an embedding medium. Please correct this statement.

Answer: *We are sorry for it, and we have corrected it in the revised manuscript (please refer to page 18 line 544).*

4. On page 18, description of the OIR model, a known region of the retina does not develop blood vessels due to exposure to hyperoxia, and upon return to normoxia, neovascularization occurs in a defined region of the tissue. As such, I recommend the analysis of the vascularization and neovascularization be limited to these areas of the tissue, and representative high resolution zoomed in images of these areas be presented in the corresponding results.

Answer: *As recommended by Reviewer #2, we have quantified the neovascularization in a defined region of retina and included zoomed images in the revised manuscript (please refer to Figs. 2D, 3D, and 5D).*

5. Authors should provide conclusive evidence supporting the idea that the effects of IL-33 siRNA were because of its effect on the retinal vascular cells and not any other cell type in the retina.

Answer: *We never meant that the effect of IL-33 siRNA is only in the retinal vascular cells, and if the reviewers felt like that, then we are sorry for our mistake. We only performed the experiment presented in Fig. 2A to assess whether our intravitreal delivery of siRNA is reaching retinal vascular cells or not? We have now included a line in the revised manuscript to refer it correctly (please refer to page 6, lines 181-182).*

6. Authors write on page 6 of the methods that "OIR model is a well-characterized VEGFA-dependent model and suppression of VEGFA expression or activity inhibits neovascularization in several diseases", but, in the opening paragraph of the results authors write " Current anti-angiogenic therapies are focused mostly on VEGFA neutralization or blockade of it signaling. However, several patients suffering from age-related macular degeneration or cancer do not respond to

anti-VEGFA therapies”. One wonders what the point of this study is if it anyway targets VEGF which is already shown to be of limited efficacy?

Answer: *We have now corrected it in the revised manuscript (please refer to page 7, lines 194-196).*

Answers to Reviewer #3:

1. Abstract: the abstract is unclear at some points because of a vague description of methods used. It would be more appealing to the reader to better understand in which models in vivo or in vitro the findings were made.

Answer: *In response to Reviewer #3's suggestions, we have now modified the abstract in the revised manuscript.*

2. Introduction: It will be interesting to explain why IL-33 was selected out of all other cytokines and interleukins.

Answer: *In response to Reviewer #3's suggestions, we have now re-written the section in the revised manuscript to include a justification (please refer to page 4, lines 101-104).*

3. Results, 2nd paragraph: authors can describe more about Fig1B and C, because there seems to be an optimum at 72 hours, whereas IL-33 levels are back to normal at 120h.

Answer: *We have modified and included a description in the revised manuscript (please refer to page 5, lines 151-156).*

4. It is unclear what the term “angiogenic events” means, instead use “angiogenic effects” or “angiogenic properties”.

Answer: *We have corrected it in the revised manuscript.*

5. Fig2A shows possible colocalization of siRNA against IL-33 and ECs (CD31). However, this is difficult to see from a 2D projection. It cannot be excluded that a supporting cell type (e.g pericyte) contains the siRNA. This can be better visualized by showing also the z-axis. Furthermore, it is also clear that other cells took up the siRNA. Please mention this in the results. After all, what is the importance of EC-specific knockdown? Isn't IL-33 secreted and then it should not matter what cells express it?

Answer: *We never meant that the effect of IL-33 siRNA is only in the retinal vascular cells, and if the reviewers felt like that, then we are sorry for our mistake. We only performed the experiment presented in Fig. 2A to assess whether our intravitreal delivery of siRNA is reaching retinal vascular cells or not? We have now included a line in the revised manuscript to refer it correctly (please refer to page 6, lines 181-182). We totally agree with the reviewer that IL-33 is secreted in the extracellular space, but studies have also found that IL-33 reside in the nucleus of cells that express them (Immunity 2005; 23, 479–490). Therefore, we used endothelial cell specific knockout mouse models.*

6. The 60% reduction in protein levels should also be presented in a bar graph or do not mention 60%.

Answer: *We have now removed it from the revised manuscript.*

7. "pathological aberrant" is a pleonasm
Answer: *Corrected it in the revised manuscript.*
8. Explain what is an "alarmin"
Answer: *It is now included in the revised manuscript (please refer to lines 217-218).*
9. Page7 "This fact can be very important": change fact to finding.
Answer: *Corrected it.*
10. Fig 4 shows the rationale for looking for localization of IL-33 siRNA in ECs, since induction of IL-33 is only happening in ECs. Thus, for a logical storyline, Fig4 would better fit prior to figure 2A.
Answer: *Please refer our answer to comment 5.*
11. Did the authors see any differences in germ line deletion versus conditional knockout of IL-33 in the OIR experiments? Please describe in Results and address this in the Discussion.
Answer: *We were unable to observe and any differences in germ line deletion versus conditional knockout of IL-33 in the OIR experiments and we have included it in the revised manuscript (please refer to page 8, lines 242-244; page 11, lines 338-342).*
12. The title of Fig7 "Jagged1 dependent Notch1 activation mediates IL-33-induced angiogenic events" is scientifically incorrect, as it has not been shown that knockdown of jagged1 leads to less notch activation.
Answer: *It is now corrected in the revised manuscript.*

We have revised and addressed all the reviewers' suggestions.

REVIEWERS' COMMENTS:

Reviewer #2 (Remarks to the Author):

The article has greatly improved following the revision. I have no additional comments. Congratulations to the authors for the Job well done revising the manuscript.

Reviewer #3 (Remarks to the Author):

The authors have addressed all issues. No further comments.